# Assessing Genetic Diversity and Searching for Selection Signatures by Comparison between the Indigenous Livni and Duroc Breeds in Local Livestock of the Central Region of Russia

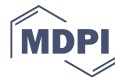

Irina Chernukha [1], Alexandra Abdelmanova [2], Elena Kotenkova [1,*], Veronika Kharzinova [2] and Natalia A. Zinovieva [2]

[1] V.M. Gorbatov Federal Research Center for Food Systems, Experimental Clinic and Research Laboratory for Bioactive Substances of Animal Origin, Talalikhina St., 26, 109316 Moscow, Russia
[2] L.K. Ernst Federal Research Center for Animal Husbandry, 142132 Podolsk, Russia
* Correspondence: lazovlena92@yandex.ru; Tel.: +79-031684478

**Abstract:** Indigenous pig breeds are mainly associated with the adaptive capacity that is necessary to respond adequately to climate change, food security, and livelihood needs, and natural resources conservation. Livni pigs are an indigenous fat-type breed farmed in a single farm in the Orel region and located in the Central European part of the Russian Federation. To determine the genomic regions and genes that are affected by artificial selection, we conducted the comparative study of two pig breeds with different breeding histories and breeding objectives, i.e., the native fat-type Livni and meat-type Duroc breeds using the Porcine GGP HD BeadChip, which contains ~80,000 SNPs. To check the Livni pigs for possible admixture, the Landrace and the Large White breeds were included into the study of genetic diversity as these breeds participated in the formation of the Livni pigs. We observed the highest level of genetic diversity in Livni pigs compared to commercial breeds ($_uH_E$ = 0.409 vs. 0.319–0.359, $p < 0.001$; $A_R$ = 1.995 vs. 1.894–1.964, $p < 0.001$). A slight excess of heterozygotes was found in all of the breeds. We identified 291 candidate genes, which were localized within the regions under putative selection, including 22 and 228 genes, which were specific for Livni and Duroc breeds, respectively, and 41 genes common for both breeds. A detailed analysis of the molecular functions identified the genes, which were related to the formation of meat and fat traits, and adaptation to environmental stress, including extreme temperatures, which were different between breeds. Our research results are useful for conservation and sustainable breeding of Livni breed, which shows a high level of genetic diversity. This makes Livni one of the valuable national pig genetic resources.

**Keywords:** Livni breed; animal genetic resources; SNPs; inbreeding; adaptation; traits

## 1. Introduction

Modern industrial pork production is based on using highly specialized, commercial pig breeds, with the most widely used being the Large White, Yorkshire, Landrace, Duroc, and Piétrain breeds. Maintaining sufficient diversity of animal genetic resources is necessary to ensure the sustainability of agricultural production, especially in conditions of climate change [1–4]. The global increase in meat consumption has led to a scenario of a high input–high output system based on sustainable intensification, maximizing animal protein production efficiency on a limited land surface at the same time as minimizing environmental impacts. The way for improving the sustainability is a reduced input–reduced output system based on selecting animals that are more robust to climate change and are better adapted to transform low quality feed (local feeds, feedstuff co-products, and food waste) into meat, but it is feasible if this is combined with a reduction in meat consumption and food waste, as well as maintaining farm income on potentially reduced production yields and reduced production efficiency by compensation through price premiums on the

products and/or savings on external inputs [5]. However, the aims of global pig production systems include a reduction in their environmental impacts, which can be achieved through an increase in outputs whilst minimizing inputs [6].

Among the FAO's description of five main breeding strategies (2010), there is a special emphasis on using local breeds, as well as improving adaptation traits of international breeds or the production traits of local breeds [7]. Therefore, local breeds are especially studied nowadays [8–11]; however, the study of commercial breeds could differ depending on growing region. Significant genetic variation has been found in Danish Duroc; the posterior mean of the additive genetic variance and heritability was found to increase in extreme environments [12]. The Chinese Duroc was found to have lost a considerable amount of genetic diversity over the past decade, while Chinese Landrace was found to have kept a high genetic diversity [13]. In another study, American Duroc had a moderate genetic differentiation with Canadian Duroc pigs, and American Duroc had more abundant genetic diversity and significantly lower level of inbreeding than Canadian Duroc pigs [14]. Thus, the region of pig farming, including such parameters as climate, feed, as well as breeding program in each country, could affect the genetic diversity of even commercial breeds.

In Russia, in the structure of the breeding stock, four breeds account for 99.2% of pigs, 53.4% of which are Large White, 21.4% are Yorkshire, 18.9% are Landrace, and 5.2% are Duroc breeds, with four local pig breeds only accounting for 0.8% of pigs [15]. The dramatic decline in the local pig genetic resources over the past few decades has been associated with targeted commercial breeding for productivity, feed conversion ratio, and carcass related traits (high lean meat content), caused by the expansion of the market for the consumption of animal products and the long-term concern of the world's population regarding the consumption of fatty meat [16–18]. A fat quality is very important in meat product processing and poor fat quality is typically assumed to result in defects and may be detrimental to processing yields [19]. The trend to lean meat production formed commercial breeds over-farming, which characterized low fat content with unacceptable properties for meat product processing. Compared to modern breeds, local pig breeds accumulate more fat, which generally contains more monounsaturated and saturated fatty acids; they exhibit a higher adipocyte size and higher activity of lipogenic enzymes [17]. Moreover, local pig breeds are mainly associated with the adaptive capacity that is necessary to respond adequately to climate change, food security, and livelihoods needs, and natural resource conservation.

Livni is one of the Russian local pig breeds that is characterized by excellent meat and fat quality. The creation of the Livni breed of pigs was started back in the first half of the 19th century by reproductive crossing of crossbreeds of different generations obtained by local long-eared pigs bred in the territory of the modern Orel and Lipetsk regions with Medium and Large White, Yorkshire and Berkshire pigs, etc. This was followed by improving the Landrace breed, which was approved in 1949. Pigs from the Livni breed are large, white, black-mottled, black and red, and give lard and meat products, and their productivity corresponds to that of Large White pigs. The Livni breed is used both for raw pork production and processed products such as minced meat and sausages. Furthermore, this breed gives a high percentage of both meat and backfat and this distinguishes it from all others. At present, only a small population of Livni pigs (458 heads [20]) is being kept in a single farm in the Oryol region. These pigs are kept in the only one gene pool farm where breeding by lines and families is used to maintain the genetic diversity and integrity. The issue is with such heterozygosity explained by the selection strategy applied for the Livni breed, because this strategy is quite different compared to breeding with commercial breeds. Moreover, 10 years ago, these Livni pigs were on free-range rearing and that had changed only due to veterinary restrictions associated with the risk of disease transmission from wild boar. Thus, the level of selection pressure in the Livni breed is much lower in comparison with others. The economic significance of this valuable local breed is based on feeding and maintaining tolerance and higher stress resistance in comparison with commercial breeds [21], as well as high meat and fat quality. The phylogenetic studies

performed using genome-wide SNPs [22] and polymorphisms of mtDNA [23] showed that the Livni breed carries the unique genetic components, which distinguish it from other Russian local and commercial pig breeds.

The aim of our work was to determine the genomic regions and genes that are affected by putative selection in the genomes of two pig breeds with different breeding history and breeding objectives, i.e., the native fat-type Livni pigs, kept as a small single local population under low selection pressure and the meat-type Duroc breed subjected to the high-throughput industrial production.

## 2. Materials and Methods

### 2.1. Samples and Genotyping

Samples (ear tissue) of Livni (LV, $n = 35$) and Duroc (DU, $n = 50$) pigs were used for the study. Additionally, since the Large White (LW) and Landrace (LN) breeds participated in the creation of the LV pigs, to characterize the genetic background of the modern population of the LV breed and to check it for the possible admixture of the LW and LN breeds, we included the last two breeds into the study.

In our study, only purebred animals of DU, LN, and LW breeds were selected. Animals of these breeds are registered in herdbook, the origin of which is confirmed by both the pedigree data and DNA analysis. For genotyping, in the case of these breeds, we selected the most unrelated individuals.

To date, the total number of LV pigs, for which the official records are collected, accounted 458 heads. However, the analysis of the pedigrees of the registered LV pigs showed that most of them had ancestors of the LN breed. We carefully checked the pedigrees of all registered LV pigs and were able to identify only about 50 animals without LN in the 1st-3rd ancestral rows. In addition, some purebred animals were closely related (siblings and half-sibs) and for this reason were excluded from the analysis. Taking into account the above arguments, we assumed that the selection of purebred non-closely related animals for our studies was the best way to obtain reliable data on the signature of selection on the genome of the LV pigs.

Samples of all breeds were sent to the laboratory of the Ernst Federal Research Center for Animal Husbandry by the private owners of the commercial breeding farms. A parentage and breed assignment of those breeds were confirmed based on the microsatellites in the laboratory of the Ernst Federal Research Center for Animal Husbandry, which has a certificate of 2020–2021 ISAG Pig STR Comparison Test (2020–2021) and has a special license accredited by the Russian Ministry of Agriculture. Commercial breeding farms and the Ernst Federal Research Center for Animal Husbandry collaborate based on the contracts. In the contract, a clause states the consent of the owners (breeding farms) to use the samples with research purpose.

Moreover, the study does not involve any endangered or protected animals and all procedures were conducted according to the ethical guidelines of the L.K. Ernst Federal Science Center for Animal Husbandry. The Commission on the Ethics of Animal Experiments of the L.K. Ernst Federal Science Center for Animal Husbandry approved the protocol No. 6 of 10 May 2021. The ear tissues were collected by trained personnel under strict veterinary rules in accordance with the rules for conducting laboratory research (tests) in the implementation of the veterinary control (supervision) approved by Council Decision Eurasian Economic Commission № 80 (10 November 2017).

Genomic DNA was extracted using the DNA Extran 2 kit (ZAO Sintol, Moscow, Russia) according to the manufacturer's instructions. Concentrations of dsDNA solutions were determined using a Qubit 1.0 fluorometer (Invitrogen, Life Technologies, Waltham, MA, USA). The OD260/280 ratio was determined using NanoDrop 2000 (Thermo Fisher Scientific, Waltham, MA, USA).

The genome-wide SNP genotyping was carried out on iScan microarray scanner (Illumina Inc., Singapore) using the Porcine GGP HD BeadChip (Illumina Inc., San Diego, CA, USA), which contains ~80,000 SNPs. In our study, we used all the capital equipment

required for SNP genotyping by Illumina SNP arrays. The equipment belongs to the Center for Collective Use "Bioresources and Bioengineering of Agricultural Animals" of the Ernst Federal Research Center for Animal Husbandry (https://www.vij.ru/infrastruktura/ckp, accessed on 10 May 2021). The SNPs genotypes of LW (*n* = 50) and LN (*n* = 50) breeds were included in the data set.

### 2.2. Quality Control

Using PLINK 1.9 software [24,25], SNP quality control was performed. All samples were subjected to filtering for genotyping efficiency (–mind 0.2). The SNPs genotyped in less than 90% of the samples (–geno 0.1) and those located on sex chromosomes were excluded from the analysis. For LV breed, kinship coefficient (relatedness) was calculated based on pair-wise identity by state matrix. The threshold for relatedness coefficient was set on 0.45. All animals of the LV breed passed through the quality control step for the kinship. We did not apply the kinship filter for commercial breeds. The final data set used for analysis included 53,263 autosomal SNPs. Additional filters for linkage disequilibrium, LD (–indep-pairwise 50.5 0.5), and minor allele frequency, MAF (–maf 0.05), were used to calculate genetic diversity, principal component analysis (PCA), Neighbor-Net tree construction, and admixture clustering that resulted in 24,861 SNPs.

### 2.3. Genetic Diversity, PCA, Neighbor-Net and Admixture

To assess the within-population genetic diversity, the observed ($H_O$) and unbiased expected ($_uH_E$) heterozygosity, the rarefied allelic richness ($A_R$), and the unbiased inbreeding coefficient ($_uF_{IS}$) were estimated using the R package, diversity [26]. Additionally, we computed the genomic inbreeding coefficient based on ROH ($F_{ROH}$) as the ratio of the sum of the length of all ROHs per animal to the total autosomal SNP coverage; for ROH estimation, see Section 2.4.2. "Runs of Homozygosity Estimation" below). PCA was performed using PLINK v1.9 software. An R package, ggplot2, was used to visualize the results [27]. Pairwise $F_{ST}$ values [28] were calculated in the R package, diveRsity [26], and used for the construction of the Neighbor-Net tree in SplitsTree software (version 4.14.5) [29]. Admixture software (version 1.3.0) [30] was employed for genetic admixture analysis and an R package, pophelper [31], was used for plotting the results. A cross-validation (CV) procedure was used to calculate the number of ancestral populations (k) from one to eight using Admixture software (version 1.3.0).

### 2.4. Selection Signature Analysis

Three different statistics were used for detecting the signatures of selection in the genome of pigs: the calculation of $F_{ST}$ values for each SNP when comparing pairs of breeds, the estimation of the ROH islands, which were overlapped among different animals within each breed, and hapFLK analysis. The search for the signatures of selection was carried out using the entire data set, comprising four breeds. The detailed analysis was performed for genomic regions identified in Livni and Duroc breeds.

#### 2.4.1. $F_{ST}$ analysis

$F_{ST}$ values [28] for all SNPs were estimated for pairs of breeds using PLINK 1.9. The top SNPs corresponding to 0.1% of $F_{ST}$ values were used to represent a selection signature, according to Kijas et al. [32] and Zhao et al. [33].

#### 2.4.2. Runs of Homozygosity Estimation

Runs of homozygosity (ROH) were detected according to the window-free method for consecutive SNP-based detection [34] using the R package, detectRUNS. One SNP with a missing genotype and up to one possible heterozygous genotype in one run were allowed to avoid the underestimation of the number of ROHs that were longer than 8 Mb [35]. The minimum ROH length was set to 500 kb for excluding the common ROHs. For minimizing false-positive results, the minimum number of SNPs (l) was calculated as it was proposed

by Lencz et al. [36] and later modified by Purfield et al. [37]. In our study, the minimum number of SNPs was 23.

Putative ROH islands were defined as overlapping homozygous regions in analyzed individuals within each breed. A threshold of 50% (the minimum proportion of animals within the breed in which overlapping ROH were detected) was selected for the LV breed, as this was suggested in other studies [38,39]. A threshold of 70% was set for the DU breed characterized by a higher level of inbreeding. We applied the threshold of 0.3 Mb for the minimal overlapping length size.

### 2.4.3. HapFLK Analysis

In this study, a hapFLK analysis was performed to detect the selection signatures through haplotype differentiation among the studied breeds using hapFLK software (version 1.4) [40]. The number of haplotype clusters per chromosome was calculated in fast-PHASE by using cross-validation and was set to 35 [41]. For detailed analyses, the hapFLK regions containing at least one SNP with a *p*-value threshold of 0.01 ($-\log10(p) > 2$) were selected.

### 2.5. Identification of Candidate Genes

For candidate gene mining in the genomic regions under putative selection, the genomic localization of the regions as detected by three different statistics was used, i.e., the $F_{ST}$, ROH, and hapFLK methods. Regions that were overlapped and revealed by at least two different techniques were prioritized. Borders of these regions according to the 10.2 genome assembly were converted to genome assembly 11.1. Genes located on the selected regions were obtained from the Ensembl Genes Release 103 database [42] based on the *Sus scrofa* gene sequence assembly.

### 2.6. Functional Enrichment Analysis

To understand the biological functions of the candidate genes, the Database for Annotation, Visualization, and Integrated Discovery (DAVID) [43] was used for enrichment analysis. Significant annotation clusters of enriched Kyoto Encyclopedia of Genes and Genomes (KEGG) pathways and Gene Ontology were selected using an enrichment score of more than 1.3 and a *p*-value of <0.05. To learn the biological functions of annotated genes and genes not included in clusters, a comprehensive literature search including information from other species was carried out.

## 3. Results

### 3.1. Genetic Diversity, Breed Relationship and Admixture

The LV pigs were characterized by a high level of genetic diversity assessed by the levels of observed heterozygosity ($H_O = 0.413$ vs. 0.325–0.371), unbiased expected heterozygosity ($_UH_E = 0.409$ vs. 0.319–0.359) and allelic richness ($A_R = 1.995$ vs. 1.894–1.964) as compared to the commercial pig breeds (Supplementary Materials, Table S1). A slight excess of heterozygotes was found in all the studied breeds compared to the expected number of heterozygotes according to the Hardy–Weinberg equation. The value of ROH based inbreeding coefficient was the lowest in LV breed ($F_{ROH} = 0.169$) comparing to the commercial breeds ($F_{ROH} = 0.272 - 0.390$) (Supplementary Materials, Table S1).

The PCA-plot (Supplementary Materials, Figure S1a) and the neighbor-joining tree (Supplementary Materials, Figure S1b) showed the breed-specific distribution of individuals of all of the studied breeds. The calculations of CV error for the different number of clusters (from one to eight) showed that the most probable number of clusters (k) is equal to four, which corresponds to the number of studied breeds. This indicates the origin of the studied breeds from four ancestral populations. An analysis of the cluster structure at k = 3 showed the participation of ancestral populations of all three commercial breeds in the development of LV breed. At k = 4, all breeds showed their own genetic structure revealing a very low level of admixture among the studied breeds (Supplementary Materials, Figure S1c).

### 3.2. Selection Signature Detection

Fifty-eight SNPs with $F_{ST}$-values beyond the cut-off (top 0.1%) were distributed among thirteen autosomes (SSA1, SSA2, SSA4, SSA5, SSA6, SSA8, SSA9, SSA11, SSA12, SSA14, SSA15, SSA17, and SSA18). The greatest numbers of SNP were found on SSA6 (8 SNPs), SSA9 (11 SNPs), and SSA14 (9 SNPs) (Figure 1, Supplementary Materials, Table S2).

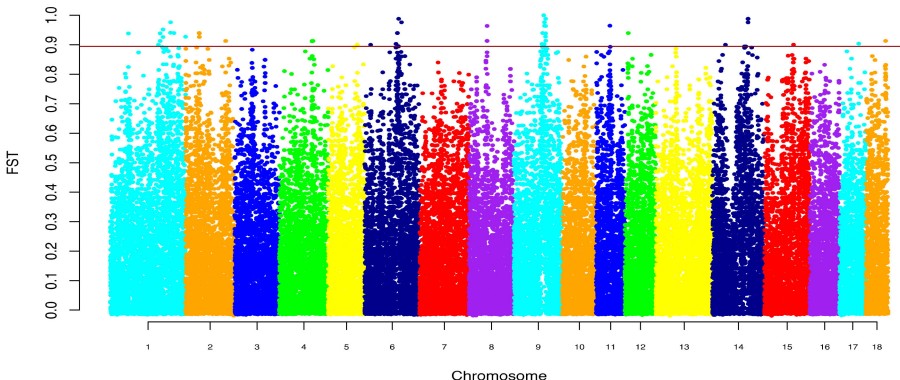

**Figure 1.** Genomic distribution of $F_{ST}$ values estimated between the LV and DU breeds. Values for the *X*-axis are pig autosomes (the breadth of autosomes corresponds to their length); and those for the *Y*-axis are $F_{ST}$ values. SNPs were plotted relative to their positions within each autosome. The threshold, which was estimated as the top 0.1% for $F_{ST}$ values, is indicated by a horizontal line. Each chromosome is marked with different color.

Thirty-four ROH islands were detected in the LV breed, which were identified in thirteen autosomes (SSA1, SSA2, SSA3, SSA4, SSA6, SSA7, SSA8, SSA11, SSA12, SSA13, SSA14, SSA15, and SSA16) and covered 33.03 Mb of the genome. Seventy ROH islands were detected in the DU breed, which were identified in fifteen autosomes (SSA1, SSA2, SSA3, SSA5, SSA6, SSA7, SSA8, SSA9, SSA10, SSA11, SSA13, SSA14, SSA15, SSA17, and SSA18) and covered 173.75 Mb of the genome (Supplementary Materials, Table S2). The number of ROH islands in the LV breed varied from 1 (SSA 6, SSA 8, SSA 12, and SSA 13) to 8 (SSA 14), and the number of ROH islands in the DU breed was from 1 (SSA 9) to 19 (SSA 1). The minima of genome coverage with ROH islands in the LV and DU breeds were detected in SSA6 and SSA8, respectively, and averaged 0.62 Mb and 0.80 Mb, and the maxima were detected in SSA 14 (9.91 Mb) and SSA 1 (42.55 Mb), respectively (Supplementary Materials, Table S3).

The hapFLK analysis resulted in the identification of 15 putative regions affected by the selection (Figure 2), including 11 regions in DU and/or LV breeds. These regions were distributed among 11 autosomes, including regions on SSA 1, SSA 3, SSA 6, and SSA 9 with a statistical significance of $p < 0.001$. The length of the putative regions under the selection pressure ranged between 0.23 and 27.94 Mb. Ten regions were breed-specific, including eight regions on SSA1, SSA3, SSA4, SSA9, SSA10, SSA12, SSA14, and SSA18 in the DU breed and two regions on SSA13 and SSA15 in LV pigs. One genomic region identified by hapFLK analysis on SSA6 (positions from 91,706,615 to 101,474,614) was common for both populations (Table 1, Supplementary Materials, Figure S2).

Comparing the genomic localization of the regions under putative selection detected by three different statistics ($F_{ST}$, ROHs, and hapFLK) revealed the presence of 15 overlapping regions, which were identified by at least two different methods (Table 2); 11 regions corresponded to the DU breed, 2 corresponded to the LV breed, and 2 were common to both studied breeds.

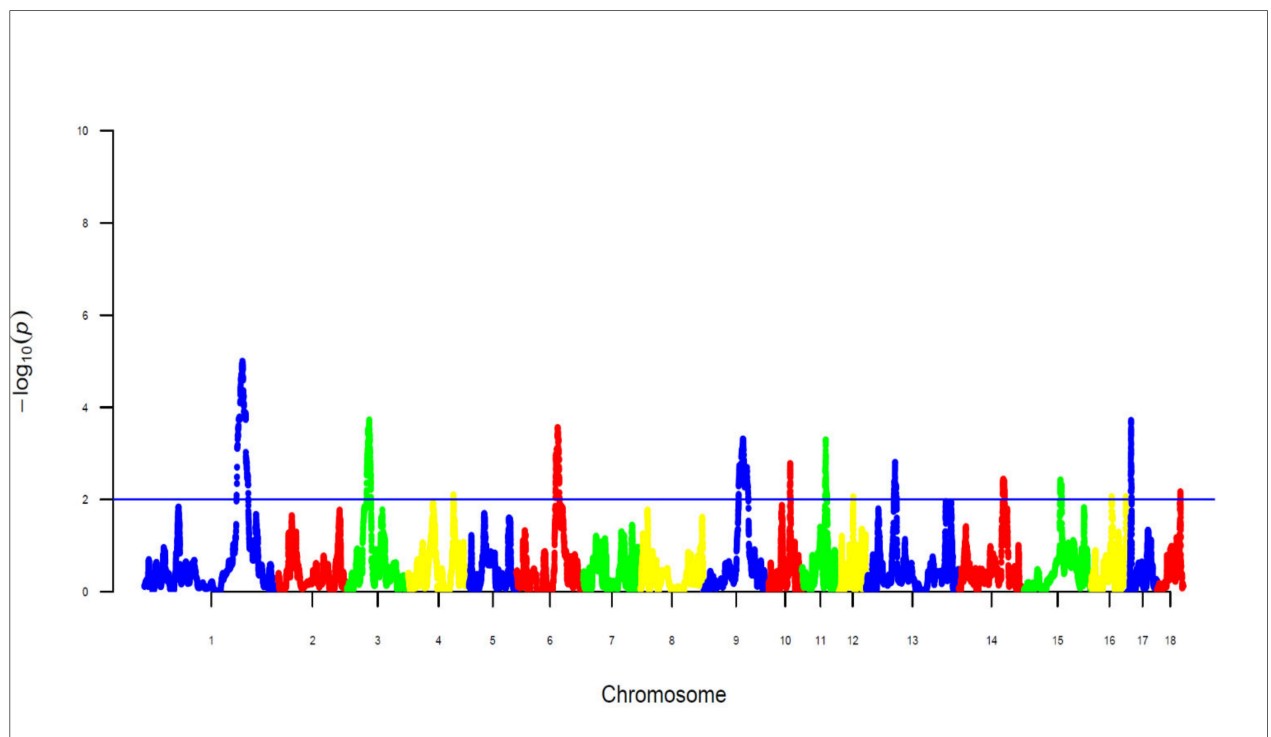

**Figure 2.** Signatures of selection in the genomes of the studied breeds based on the hapFLK statistics. Values for the *X*-axis are pig autosomes, and those for the *Y*-axis are values of statistical significance ($-\log_{10}$ *p*-values). The blue line indicates the threshold of significance at $p < 0.01$ (i.e., $-\log_{10}(p) > 2$). Each chromosome is marked with different color. Magnified plots for 11 putative regions identified by hapFLK analysis in LV and/or DU breeds are presented in Supplementary Materials, Figure S2.

**Table 1.** HapFLK regions identified in the genome of the studied LV and DU pig populations.

| SSA * | Breed | Position of Region | | Amount of SNP in Region | Length, Mb | The Most Significant SNP | *p*-Value |
|---|---|---|---|---|---|---|---|
| | | **Start** | **End** | | | | |
| 1 | DU | 216,980,027 | 244,920,837 | 255 | 27.94 | 231,073,909 | 0.000010 |
| 3 | DU | 45,778,105 | 56,709,945 | 154 | 10.93 | 52,588,363 | 0.000187 |
| 4 | DU, LV | 106,411,407 | 106,750,789 | 10 | 0.34 | 106,485,236 | 0.007915 |
| 6 | DU, LV | 91,706,615 | 101,474,614 | 134 | 9.77 | 97,214,894 | 0.000275 |
| 9 | DU | 82,692,233 | 105,612,360 | 273 | 22.92 | 93,125,776 | 0.000487 |
| 10 | DU | 49,911,155 | 51,759,133 | 61 | 1.85 | 50,699,204 | 0.001653 |
| 12 | DU | 33,122,614 | 33,348,283 | 7 | 0.23 | 33,220,015 | 0.008654 |
| 13 | LV | 65,209,239 | 71,634,446 | 68 | 6.43 | 68,207,174 | 0.001558 |
| 14 | DU | 103,168,211 | 108,016,508 | 89 | 4.85 | 104,929,965 | 0.003601 |
| 15 | LV | 84,563,546 | 88,484,722 | 81 | 3.92 | 85,635,190 | 0.003762 |
| 18 | DU | 53,956,913 | 54,852,652 | 23 | 0.90 | 54,392,049 | 0.006769 |

* SSA, *Sus scrofa* autosomes. Four breeds (DU, LV, LN, and LW) were used for the analysis; only the hapFLK regions are shown, which were identified in the LV and/or Duroc breeds.

The list of the regions under putative selection was expanded with six regions that were identified in both pig breeds by at least one of the methods (ROH, hapFLK) (Table 3).

**Table 2.** Overlapped genomic regions and/or SNPs under putative selection identified by at least two different statistics in the Duroc (DU) and Livni (LV) breeds.

| SSA * | $F_{ST}$ [a] | | ROH [b] | | hapFLK [c] | |
|---|---|---|---|---|---|---|
| | Breed | Position | Breed | Position | Breed | Position |
| 1 | DU/LV | 198,346,039 | DU | 198.0–202.1 | | |
| 1 | DU/LV | 229,476,564; 229,502,611 | DU | 228.6–231.0 | DU | 217.0–244.9 |
| 1 | DU/LV | 272,689,388; 272,760,898 | DU | 271.9–275.6 | | |
| 2 | DU/LV | 31,566,031; 32,186,193; 32,313,049; 32,319,002; 32,407,451 | DU | 31.3–33.6 | | |
| 3 | | | DU | 49.2–54.7 | DU | 45.7–56.7 |
| 4 | DU/LV | 106,698,421; 106,719,032 106,750,789 | | | DU, LV | 106.4–106.7 |
| 6 | DU/LV | 90,705,621; 91,279,252 94,442,844; 94,451,345; 94,775,420; 95,482,175 | DU | 88.0–93.2 93.5–99.2 | DU, LV | 91.7–101.5 |
| 9 | DU/LV | 82,963,397; 85,926,552; 93,596,926; 95,858,320; 97,264,389; 97,527,550; 102,510,717; 103,035,428; 103,174,861; 103,267,375; 104,996,366 | DU | 75.8–106.5 | DU | 82.7–105.6 |
| 10 | | | DU | 49.5–49.9 | DU | 49.9–51.8 |
| 14 | DU/LV | 100,350,445; 104,208,040; 104,282,405; 105,893,370; 106,938,671 | LV | 100.1–101.4 | | |
| 14 | DU/LV | 114,848,572; 114,895,388; 114,958,111 | DU | 100.1–109.3 110.0–117.8 | DU | 103.1–108.0 |
| 15 | LV/LW | 84,696,087 | LV | 84.7–85.8 | LV | 84.6–88.5 |
| 15 | LV/LN | 90,388,740 | LV | 90.4–91.4 | | |
| 15 | DU/LV | 121,814,208 | DU | 118.8–121.7 | | |
| 18 | DU/LV | 54,277,674 | DU | 54.1–55.8 | DU | 53.9–54.9 |

* SSA, *Sus scrofa* autosomes. Methods used for defining the signatures of selection: [a] $F_{ST}$, top 0.1% SNPs by the $F_{ST}$ value at pairwise population comparison (LW and LN breeds were used only for pair-wise breed comparison with LV pigs); [b] ROH, ROH segments distributed in more than 50% of animals for the LV breed and 70% for the DU breed; and [c] hapFLK, regions identified by hapFLK analysis at $p < 0.01$.

**Table 3.** Common selection signatures in the genome of pigs of the Duroc (DU) and the Livni (LV) breeds identified by the ROH or hapFLK methods.

| SSA * | The LV Breed | | The DU Breed | |
|---|---|---|---|---|
| | Method | Position | Method | Position |
| 1 | ROH | 71,814,075–72,721,133 | ROH | 71,950,726–72,721,133 |
| 1 | ROH | 83,260,076–84,223,593 | ROH | 83,260,076–84,223,593 |
| 1 | ROH | 241,903,331–242,955,813 | hapFLK | 216,980,027–244,920,837 |
| 6 | ROH | 71,436,086–72,057,699 | ROH | 71,303,189–72,477,552 |
| 11 | ROH | 34,824 047–39,790,178 | ROH | 36,953,937–40,366,928 |
| 14 | ROH | 100,097,831–101,350,552 | ROH | 100,162,325–109,285,369 |

* SSA, *Sus scrofa* autosomes. Methods used for defining the signatures of selection: *ROH*, ROH segments distributed in more than 50% of animals for the LV breed and 70% for the DU breed; and [4] *hapFLK*, regions identified by hapFLK analysis at $p < 0.01$.

### 3.3. Candidate Gene Determination

The structural annotation of these regions revealed the presence of 291 candidate genes in the two studied pig populations; 228 genes were specific to DU pigs, 22 were specific to LV pigs, and 41 were specific to both studied breeds (Table 4). Details of the gene names are presented in Supplementary Materials, Tables S4–S6.

**Table 4.** Genes within the overlapped genomic regions affected by putative selection.

| SSA * | Methods [a] | Region (Mb) | Genes [b] |
|---|---|---|---|
| | | | The LV breed |
| 15 | ROH, $F_{ST}$, hapFLK | 84.7–85.8 | *ABCB11, LRP2, DHRS9, FASTKD1, CCDC173, KLHL23, UBR3, MYO3B, GAD1, GORASP2, TLK1, METTL8, DCAF17, CYBRD1, SLC25A12, METAP1D, DLX1, ITGA6, PDK1, RAPGEF4* |
| 15 | ROH, $F_{ST}$ | 90.4–91.4 | *LNPK, HOXD13* |
| | | | The DU breed |
| 1 | ROH, $F_{ST}$ | 198.0–202.1 | *LRR1, KLHDC2, SOS2, CDKL1, MAP4K5, ATL1, SAV1, NIN, PYGL, TRIM9, TMX1, FRMD6, RTRAF, NID2* |
| 1 | ROH, $F_{ST}$, hapFLK | 217.0–244.9 | *KCNH5, TEK, U6, ELAVL2, ZEB2, CDKN2B, P14ARF, KLHL9, FOCAD, MLLT3, SLC24A2, SAXO1, ADAMTSL1, SH3GL2, CNTLN, BNC2, CCDC171, PSIP1, TTC39B, FREM1, NFIB, MPDZ, LURAP1L, PTPRD, KDM4C, GLDC, RANBP6, IL33, KIAA2026, MLANA, ERMP1, PDCD1LG2, CD274, JAK2, RCL1, SPATA6L, GLIS3, PUM3, KCNV2* |
| 1 | ROH, $F_{ST}$ | 271.9–275.6 | *GRIN3A, SMC2, OR13C3, ABCA1, NIPSNAP3A* |
| 2 | ROH, $F_{ST}$ | 31.3–33.6 | *PAX6, ELP4, IMMP1L, DCDC1, MPPED2* |
| 3 | ROH, hapFLK | 45.7–56.7 | *ZC3H8, MERTK, ACOXL, BUB1, SEPTIN10, NPHP1, ZNF514, PROM2, KCNIP3, ARID5A, NCAPH, SNRNP200, STARD7, SH3RF3, EDAR, CCDC138, RANBP2, SLC5A7, ST6GAL2, UXS1, NCK2, FHL2, GPR45, TGFBRAP1, SLC9A2, IL18RAP, IL18R1, IL1RL1, IL1R1, IL1R2, RFX8, CREG2, RNF149, CNOT11, TBC1D8, CHST10* |
| 9 | ROH, $F_{ST}$, hapFLK | 75.8–106.5 | *ZNF804B, CDK14, AKAP9, CYP51A1, ANKIB1, PEX1, CDK6, SAMD9, VPS50, CALCR, GNGT1, COL1A2, CASD1, SGCE, PPP1R9A, PON3, PON2, ASB4, PDK4, U6, DYNC1I1, SLC25A13, TAC1, ASNS, COL28A1, MIOS, UMAD1, GLCCI1, ICA1, NXPH1, CHRDL2, PHF14, THSD7A, TMEM106B, SCIN, DGKB, AGMO, MEOX2, CRPPA, TSPAN13, AGR2, AHR, SNX13, HDAC9, TMEM196, ITGB8, ABCB5, SP4, DNAH11, CDCA7L, RAPGEF5, TOMM7, KLHL7, NUP42, IGF2BP3, RUNDC3B, CROT, ELAPOR2, GRM3, SEMA3A* |
| 10 | ROH, hapFLK | 49.5–51.8 | *RSU1, C1QL3, PTER, MINDY3, ITGA8, FAM171A1, NMT2, ACBD7, SUV39H2, HSPA14* |
| 14 | ROH, $F_{ST}$, hapFLK | 101.4–109.3 | *PRKG1, A1CF* |
| 14 | ROH, $F_{ST}$ | 110.0–117.8 | *CH25H, LIPA, IFIT2, SLC16A12, PANK1, KIF20B, RPP30, PCGF5, HECTD2, BTAF1, CPEB3, MARCHF5, IDE, HHEX, EXOC6, CYP26C1, MYOF, CEP55, PDE6C, FRA10AC1, LGI1, PLCE1, NOC3L, TBC1D12, HELLS, CYP2C42, PDLIM1, SORBS1, ALDH18A1, TCTN3, ENTPD1, CC2D2B, ZNF518A, BLNK, DNTT, OPALIN, TLL2, PIK3AP1* |
| 15 | ROH, $F_{ST}$ | 118.8–121.7 | *PARD3B, NRP2, INO80D, NDUFS1, ZDBF2, ADAM23, DYTN, CPO* |
| 18 | ROH, $F_{ST}$, hapFLK | 53.9–55.8 | *CCDC201, ADCY1, CCM2, PURB, MYO1G, ZMIZ2, OGDH, DDX56, NUDCD3, GCK, CAMK2B* |
| | | | The LV and the DU breeds |
| 1 | ROH | 71.8–72.7 | *UFL1, FHL5, GPR63, KLHL32, MMS22L* |
| 1 | ROH | 83.2–84.2 | *SEC63, OSTM1, SNX3, AFG1L* |
| 1 | ROH, hapFLK | 241.9–243.0 | *KIAA2026, MLANA, ERMP1, U6, PDCD1LG2, CD274, JAK2* |
| 4 | $F_{ST}$, hapFLK | 106.4–106.7 | *RORC, TDRKH, MRPL9, TUFT1, SNX27* |
| 6 | ROH | 71.3–72.5 | *ALDH4A1, UBR4, CAPZB* |
| 6 | $F_{ST}$, hapFLK | 91.7–101.5 | *RAB31, ANKRD12, MTCL1, PTPRM, ARHGAP28, EPB41L3, MYOM1, SMCHD1, NDC80, METTL4, GREB1L, CABLES1, TMEM241, NPC1, LAMA3* |
| 11 | ROH | 34.8–40.4 | *TDRD3, SPOCK1* |
| 14 | ROH, $F_{ST}$, hapFLK | 100.1–101.4 | - |

* SSA, *Sus scrofa* autosomes. [a] Methods used for defining the signatures of selection: ROH, runs of homozygosity islands shared in more than 50% of animals for the LV breed and 70% for the DU breed; hapFLK, regions identified by hapFLK analysis at $p < 0.01$; and $F_{ST}$, top 0.1% of SNPs by $F_{ST}$ value at the pairwise population comparison. [b] Candidate genes.

All the discovered 22 candidate genes for the LV breed were found on the SSA15 chromosome, and the total length of presumptive selection footprints was 2.1 Mb. Among the discovered 228 candidate genes for the DU breed, 58 genes were found on SSA1, 5 on SSA2, 36 on SSA3, 60 on SSA9, 10 on SSA10, 40 on SSA14, 8 on SSA15, and 11 on SSA18. The discovered candidate genes were found on eight autosomes, and the total length of presumptive selection footprints was 102.5 Mb. Among the discovered 41 candidate genes for both the LV and the DU breed, 16 genes were found on SSA1, 5 on SSA4, 18 on SSA6, and 2 on SSA11. The discovered candidate genes were found on four autosomes, and the total length of presumptive selection footprints was 20.2 Mb.

*3.4. Functional Enrichment Determination*

Using the DAVID web tool and a list of 291 candidate genes found in the genomic regions with selection signatures, 214 genes with described functions were identified (Supplementary Materials, Tables S4–S6). The significant clusters are shown in Supplementary Materials, Table S7.

Among the discovered 22 candidate genes for the LV breed, biological processes were identified for 16 in GO terms [44]. However, no genes were reliably associated with a specific functional category (Supplementary Materials, Tables S4 and S7). Only GO:0016567~protein ubiquitination was enriched (*p*-value: 0.092) and included genes *DCAF17* and *KLHL23*. A detailed analysis of the molecular functions of the individual genes identified in LV pigs revealed their participation in various biological processes, including the development of the skeleton and skeletal muscles *(HOXD13)*, fat cell differentiation *(METTL8)* and brown fat cell differentiation *(ITGA6)* (Supplementary Materials, Table S4).

Among the discovered 41 candidate genes for both the LV and the DU breeds, functional annotation was performed for 28 genes in GO terms [44]. Similarly, no genes were reliably associated with a specific functional category. Genes with molecular functions linked with the WASH complex, actin assembly, and F-actin were the most represented (Supplementary Materials, Tables S5 and S7).

The IPR015621: interleukin-1 receptor family was enriched in most candidate genes in the DU pigs (*p* < 0.001). A detailed analysis of the molecular functions of the individual genes identified in DU pigs revealed their participation in the regulation of growth and development *(AGR2, CCM2)*, actin regulation processes *(FRMD6, NCK2, SCIN, SORBS1, TEK)*, amino acid biosynthesis *(ALDH18A1, ASNS)*, the biosynthesis and metabolism of fatty acids *(CH25H, CROT)*, bone formation *(FHL2)*, glucose metabolism, insulin dependence *(ADCY1, CAMK2B, IDE, GCK, NID2, PDE6C, PDK4, PAX6, PYGL, SORBS1, SOS2)*, and pigmentation *(EDAR)* (Supplementary Materials, Tables S6 and S7).

Comparing the performance characteristics of LV and DU pigs bred in Russia (based on the genetic evaluation by 2020) [20], it should be noticed that the number of LV piglets born alive is higher, as well as feed conversion rate and backfat thickness (Supplementary Materials, Tables S8 and S9).

**4. Discussion**

To elucidate the history of artificial selection of an indigenous fat-type Livni pig, we performed the genome-wide SNP-genotyping of 35 pure-bred animals which were carefully selected based on pedigree analysis and revealed no ancestors of other breeds in at least three rows of pedigrees. The meat-type Duroc breed was selected as a comparison breed because of its different breeding objectives. To check for the possible admixture, the Large White and Landrace breeds were included in the analysis of genetic diversity indices, because these two breeds participated in the formation of Livni pigs at different stages of its development.

We observed the highest level of genetic diversity and lowest level of genomic inbreeding in Livni pigs compared to commercial breeds (Supplementary Materials, Tables S1 and S10), which agreed with the other studies revealing the greatest level of genetic variability in local breeds compared to high-producing transboundary breeds [22,45–47]. This may be a

consequence of the participation of various breeds in the development of the Livni breed. Another possible reason could be the lack of strict selection pressure for a limited number of economically important traits. On the contrary, the Duroc breed revealed the lowest genetic diversity and greatest level of genomic inbreeding, which agreed with previous studies [48]. A sample of the Duroc breed belonging to the breeding nucleus that has undergone the high selection pressure for a very limited number of traits, might be reflected in the increase of autozygosity in this breed. The strong signatures of selection in the Duroc breed that can affect the lean muscle mass were previously reported [49]. The PCA plot, Neighbor-Net tree, and admixture-plot (Supplementary Materials, Figure S1a–c) revealed the own genetic background of Livni pigs that make them one of the priority objects for genomic studies.

Using three different statistics (top 0.1 $F_{ST}$ at pair-wise breed comparison, ROH islands and hapFLK analysis), we selected eleven Duroc-specific, two Livni-specific, and eight common genomic regions for detailed analysis, which were identified by at least two different methods or were found in both pig breeds by ROH or hapFLK analysis (Tables 2 and 3). Among 291 candidate genes, which were localized within selected genomic regions (Table 4), 214 genes had the described functions in GO-terms; among them, 170 genes were found within genomic regions, identified in the Duroc breed, 16 genes—in the Livni breed, and 28 genes—in both of the breeds (Supplementary Materials, Tables S4–S6).

In the Duroc breed, a large number of candidate genes is involved in glucose and lipid metabolism, in muscle fiber formation, and meat quality traits. According to GO analysis, PDK4 is strongly involved in both lipid and glucose metabolism. Moreover, it was determined in Nanyang black pigs to be a transcription or translation regulator of lipid deposition genetic divergence [50]. DGKB is involved in fat deposition in pigs [51]. PIK3AP1 was found to be more highly expressed in fat pigs than in lean ones [52]. Genetic variants mapped to FHL2 modulate lipid metabolism and control energy homeostasis in pigs [53]. ELP4 influences backfat thickness in Yorkshire pigs [54]. RUNDC3B is considered a backfat gene [55]. GCK is upregulated in pigs with higher backfat thickness and is involved in fatty acid synthesis [56]. Interestingly, although ZMIZ2 is associated with the "ease of waking up in the morning" [57], it also could be linked with backfat thickness [58]. Another group of genes is also associated with lipid metabolism or alterations (ADAM23 [59], AGR2 [60], CH25H, CYP2C42, CYP51A1 [61]), adipogenesis (CDCA7L [62], CDK14 [63], NFIBs [64], NID2 [65], TAC1 [66], SH3RF3 [67]), or adipose tissue deposition and accumulation in certain localizations (CDKN2B [68], ASB4 [69], BTAF1 [70], NID2 [71], THSD7A [72]). The SORBS1 gene plays a key role in adipogenesis [73]; the expression level of SORBS1 was increased in Wei pigs and could affect fat deposition in muscles [74]. PTPRD influences meat quality, including pork [11,75]. ALDH18A1 is positively correlated with lean meat and bone traits [76] and FOCAD is linked with body weight and lean mass [77]. A significant association between TGFBRAP1 and meat quality traits was also revealed in DU pigs [78]. FRMD6, ITGA8, ITGB8, MYO1G, KLHDC2 [79], and SMC2 [80] genes are associated with growth and meat production traits, whereas CALCR is involved in osteoclast differentiation and associated with body conformation traits [81]. These genes could be important in muscle and fat tissue formation in pigs, including VAT, SAT, and IMF, and thus influence the quality of meat raw materials.

Some genes associated with adaptation and immunity were also found in DU pigs. IL1R2, as a candidate gene, was detected in Chinese pigs, which might explain the high resistance to disease in these pigs [82]. IL33 maintains immune homeostasis in adipose tissue [83], activates type 2 immune responses, and licenses brown and beige adipocytes for uncoupled respiration. In the absence of IL-33, beige and brown adipocytes develop normally but fail to express an appropriately spliced form of UCP1 mRNA, resulting in the absence of UCP1 protein and impairment in uncoupled respiration and thermoregulation, which is especially important during the perinatal period [84]. On the other hand, pigs do not have the UCP-1 isoform [85,86] as well as brown adipocytes, but could have beige ones, that are very important for thermoregulation. Beige adipocytes were found in inguinal

subcutaneous WAT, axillary sWAT, and perirenal fat from acute cold-stimulated cold-tolerant pig breeds in China, including Tibetan pigs and Min pigs [87]. Differentiated beige cells were also observed in the subcutaneous fat of Tibetan pigs [88]. CDK6 regulates beige adipocyte formation; its kinase activity negatively regulates the conversion of fat-storing cells into fat-burning cells [89]. The role of LIPA in lipid metabolism is evident. Moreover, LIPA regulates fatty acid channeling in brown adipose (BAT) tissue to maintain thermogenesis and is critical for shuttling fatty acids derived from circulating lipoproteins to BAT during cold exposure [90]. It has been reported that the expression of NDUFS1 is higher in brown adipocytes than in white adipocytes [91]. Recent studies have shown that P14ARF regulates adipose tissue (AT) physiology and adipocyte functions such as lipid storage, inflammation, oxidative activity, and cellular plasticity (browning) [92]. Another study reported that OGDH is also involved in heat production in beige/brown adipose tissue and was identified to be an up-regulated gene to induce the browning of SC-WAT, without any cold stimulation or fasting [93]. However, cold exposure could also up-regulate this gene [94]. PANK1 is also involved in the beige differentiation program [93], is enriched in BAT compared with WAT, is cold-regulated, and is associated with the brightening of WAT [95]. TBC1D12 is linked to the signatures of adaptations, caused by temperature and sunlight [96,97], and is a strong candidate gene for selection in response to environmental stress [98] and climatic selection [4]. Positive selection in the ZDBF2 gene is associated with some yet known physiological or immunological adaptations of animals to low temperatures during the adaptation to Arctic or Antarctic environments [99,100]. ZDBF2 appears to be a new paternally expressed growth-promoting gene and tunes the control of feeding and growth in neonates [101,102]. ZNF804B is associated with selective adaptation in cattle and can be important for local adaptation in sheep [103]. The genes associated with adaptation, especially to cold, are very important for sustainable pig farming, especially in the case of the local environment conditions of growing in the central region of Russia.

Genomic regions, which are common for two breeds, include the genes which are involved in adipogenesis (METTL4 [104], SPOCK1 [105], JAK–STAT [106]), immunity (PDCD1LG2 [107], UBR4 [108]), the formation of meat traits, and muscle fiber (ALDH4A1 [73], RORC [109,110], TDRKH [111], CAPZB [112]).

Among 22 Livni breed-specific genes, we found the candidate genes which could be linked with muscle and skeletal muscle formation, adipogenesis, oxidative stress, and glucose and insulin metabolism. It has been reported that DHRS9 is highly induced during cold exposure [113], while LRP2 has been shown to be a critical node in the hypothalamic control of energy metabolism [114]. SLC25A12 has been shown to be significantly up-regulated in BAT and to be sensitive to cold exposure, but not significantly [115,116]. TLK1 is proposed to be important for adipogenesis [117,118]. According to GO analysis, METTL8 is involved in both skeletal muscle tissue development and fat cell differentiation. Prior studies have linked METTL8 to adipogenesis [119,120], but this gene has also been associated with meat quality traits [121]. MYO3B is encoded with class III myosin B [122]. Interestingly, ITGA6 is involved in numerous important processes, including nail, skin, and renal system development, the regulation of actin cytoskeleton, and brown fat cell differentiation. ITGA6, which encodes the major ITGA in adipocytes, is highly expressed in subcutaneous fat tissue and in intramuscular fat [123]. HOXD13 participates in the skeletal system and male genitalia development, according to GO analysis.

Due to the diversification of farming systems and climate change, farm animals are exposed to environmental disturbances to which they respond differently depending on their robustness [124]. The definition of the breeding objectives as well as evaluation of genetic merit needs to be based on the local environment [125]. High genetic variation is related to higher adaptive potential to new environmental conditions such as changing climate conditions or the emergence of new pathogens [126]. The discovered genes could be important in muscle and fat tissue formation in pigs, including VAT, SAT, and IMF, and thus influence the quality of meat raw materials. The genes associated with adaptation,

especially to cold, are very important for sustainable pig farming, especially in the case of the local environment conditions of growing in central region of Russia.

## 5. Conclusions

Genetic diversity is very important for pig farming sustainability, especially for local production systems. Indigenous breeds are essential genetic resources due to their good adaptability to the local environmental conditions. Interestingly, commercial breeds could also be genetically sensitive to climate change, feed quality, and other housing and environment factors during long periods of time after being placed in certain geographic regions. Comparative genomic studies of two breeds with a different history of artificial selection and breeding objectives, i.e., fat-type native Livni breed and meat-type commercial Duroc breed, revealed only a small part of common signatures of selection. Among 214 genes with described functions in GO-terms, 170 and 16 genes were specific for Duroc and Livni breeds, respectively, while only 28 genes were common for both of the breeds. We identified the candidate genes associated with carcass related traits, including fat and meat traits, as well as with adaptation capacity, including cold tolerance. The interleukin-1 receptor family was enriched in Duroc pigs, while protein ubiquitination was enriched in the Livni breed. Our research results are significant for pig farming sustainability in Russia, taking into account specific local environment conditions and the nutritive potential of pork occupying 35% of the total meat consumption in the country.

**Supplementary Materials:** The following are available online at https://www.mdpi.com/article/10.3390/d14100859/s1, Figure S1: Genetic relationships between the Livni, Duroc, Landrace, and Large White breeds based on PCA (a), Neighbor-Net tree (b) and admixture clustering (c); Figure S2: Magnified plots for 11 putative regions identified by hapFLK analysis in LV and/or DU breeds; Table S1: Summary of genetic diversity statistics calculated for the studied pig breeds; Table S2: Summary of the selective sweeps and candidate SNPs observed in the genome of Livni and Duroc breeds; Table S3: The distribution of ROH island number and length in chromosomes; Table S4: Functional annotation and enrichment of Gene Ontology (GO) terms among the identified genes within the sweep regions found in the LV breed as ascertained by DAVID; Table S5: Functional annotation and enrichment of Gene Ontology (GO) terms among the identified genes within the sweep regions found in LV and DU breeds as ascertained by DAVID; Table S6: Functional annotation and enrichment of Gene Ontology (GO) terms among the identified genes within the sweep regions found in the DU breed as ascertained by DAVID; Table S7: Functional Gene Ontology (GO) terms enriched with candidate genes; Table S8: Productivity of Livni pigs comparing to Duroc pigs bred in Russia (based on the genetic evaluation by 2020); Table S9: Growth, feed efficiency and carcass traits; Table S10: Individual genomic inbreeding coefficients calculated based on ROHs.

**Author Contributions:** Conceptualization, I.C. and N.A.Z.; methodology, I.C., N.A.Z., A.A. and E.K.; software, A.A.; validation, A.A. and E.K.; formal analysis, V.K.; investigation, V.K.; resources, I.C. and N.A.Z.; data curation, A.A.; writing—original draft preparation, I.C. and E.K.; writing—review and editing, I.C. and N.A.Z.; visualization, A.A.; supervision, I.C. and N.A.Z.; project administration, I.C.; funding acquisition, I.C. All authors have read and agreed to the published version of the manuscript.

**Funding:** The study was carried out within project No. 21-76-20032, supported by the Russian Science Foundation. The genotypes of Large White and Landrace breeds were produced with the financial support of the Russian Ministry of Science and Higher Education within theme No. FGGN-2022-0002.

**Institutional Review Board Statement:** The study was conducted according to the guidelines of the Declaration of Helsinki and the ethical guidelines of the L.K. Ernst Federal Research Center for Animal Husbandry. Protocol No. 6 was approved by the Commission on the Ethics of Animal Experiments on 10 May 2021.

**Data Availability Statement:** The genotyping data presented in this study can be shared with the third parties upon reasonable request.

**Conflicts of Interest:** The authors declare no conflict of interest.

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
