# Peer review of "Assessing Genetic Diversity and Searching for Selection Signatures by Comparison between the Indigenous Livni and Duroc Breeds in Local Livestock of the Central Region of Russia"

_diversity, doi:10.3390/d14100859_

Round 1
Reviewer 1 Report
Chernukha et al. assessed the genetic diversity of the indigenous Livni pig breeds from the central region of Russia using the SNPs. They reported a high level of genetic diversity in the Livni breed and claimed no ancestors of other breeds in at least three rows of pedigrees of the Livni breed. They have followed the standard molecular procedure of laboratory analysis and the data analysis also seems acceptable. The major issue I find in the manuscript is in the write-up. I suggest authors to shorten the manuscript by keeping only relevant information and maintaining the flow of the information.
Comments:
The abstract fails to stand alone. I suggest authors to rewrite it focusing on their results and conclusion.
The major concern is in the write-up. For example, in the very first paragraph of the Introduction section, the authors have raised the context, and provided some facts with examples but failed to conclude. Every paragraph suffers the same.
Research questions are not clearly defined. What could be the means of such artificial selections? Need to discuss.
What knowledge does the MS add beyond Transpov et al. 2016, (https://doi.org/10.1186/s12711-016-0196-y); Kharzinova et al. 2022
(https://doi.org/10.1134/S102279542208004X)? I could not see these papers discussed in the manuscript. Were they intentionally missed?
The manuscript title has inconsistency in typology.
Line 20: ………………0.319-0,359……….. please replace the comma with a decimal.
Lines 35-36: ……The way for improving sustainability is a reduced input–reduced output system based on selecting animals…. Is this true? How can a reduced output be sustainable?
Lines 40-41: ………however, commercial breeds could be differ depending on growing region………….. Please fix the grammatical error.
Lines 42-43: It was found significant genetic variance………….. Please change it to- Significant genetic variation was found ……………..
Lines 64-65: Please rephrase
Line 127: Flurometer was used to check ‘purity’ of DNA? OR concentration?
Line 133: Large White and Landrace… Already abbreviated, use abbreviation.
Line 155: ……….ancestral populations (K) from one to five…………How was this range determined? Results says the range of 0ne to eight. Please check it.
Be consistent in writing the term FST.
Line 215: CV error- This abbreviation has never been defined prior to this.
Line 216: … from one to eight…. Conflicted with methods. Line 155
Tables: Better to use other symbols than 1, 2. For example, n2 looks like square of n.
Table 5: The details of the gene names?
I would love to read it again if the authors consider shortening the manuscript retaining the important message they want to deliver.
Author Response
We would like to thank the Reviewer 1 for time spent carefully reading of our article. All the given comments, which have been very helpful in improving our manuscript, have been taken into account. Please, find below our response to the comments. We marked all of changes in the manuscript by red.
Comment: Chernukha et al. assessed the genetic diversity of the indigenous Livni pig breeds from the central region of Russia using the SNPs. They reported a high level of genetic diversity in the Livni breed and claimed no ancestors of other breeds in at least three rows of pedigrees of the Livni breed. They have followed the standard molecular procedure of laboratory analysis and the data analysis also seems acceptable. The major issue I find in the manuscript is in the write-up. I suggest authors to shorten the manuscript by keeping only relevant information and maintaining the flow of the information. I would love to read it again if the authors consider shortening the manuscript retaining the important message they want to deliver.
Respond: Dear Reviewer, we are very grateful for your positive feedback on our work. We’ve shortened the manuscript by moving Tables 1 and 6 in Supplementary Materials and reducing the Discussion section
Comment: the abstract fails to stand alone. I suggest authors to rewrite it focusing on their results and conclusion.
Response: we modified the abstract, highlighting, that the local pig breeds are the valuable source of genetic diversity, that make them capable to respond adequately to climate change, food security and livelihoods needs.
Comment: The major concern is in the write-up. For example, in the very first paragraph of the Introduction section, the authors have raised the context, and provided some facts with examples but failed to conclude. Every paragraph suffers the same.
Response: we rewrote the introduction and added the small conclusions after each paragraph in the Introduction section.
Comment: Research questions are not clearly defined. What could be the means of such artificial selections? Need to discuss.
Response: we made corrections in the Introduction section. Additionally, we clarified the aim of our research, highlighting the differences in breeding objectives of Livni and Duroc breeds, as well as in production systems used for breeding of studied breeds.
Comment: What knowledge does the MS add beyond Transpov et al. 2016, (https://doi.org/10.1186/s12711-016-0196-y); Kharzinova et al. 2022
(https://doi.org/10.1134/S102279542208004X)? I could not see these papers discussed in the manuscript. Were they intentionally missed?
Response: Transpov et al. [2016] considered genetic distances, population structures, genetic diversity and demographic history of pig breeds from Russia, Belorussia, Kazakhstan and Ukraine, including the sample Livni breed of small sample size (n=16). Kharzinova et al. [2022] also considered genetic diversity and phylogenetic relationships of Russian pig breeds, using the analysis of mtDNA D-Loop polymorphism. The search for the signature of selection in Livni pigs was not conducted in that studies. In the present manuscript, we focused on searching the putative genomic regions and genes that are affected by artificial selection by comparing the local fat-type Livni breed with high-producing commercial meat-type Duroc breed. We mentioned the both of the manuscript in the Introduction section and added to the References (see ref. 22 and ref. 23). We also cited ref. 22 in Discussion section in connection with higher genetic diversity observed in local breeds comparing to commercial ones.
Comment: The manuscript title has inconsistency in typology.
Response: We’ve changed the title.
Comment: Line 20: ………………0.319-0,359……….. please replace the comma with a decimal.
Response: Corrected.
Comment: Lines 35-36: ……The way for improving sustainability is a reduced input–reduced output system based on selecting animals…. Is this true? How can a reduced output be sustainable?
Response: We thank Reviewer for comment. We’ve rewritten this part of the Introduction section.
Comment: Lines 40-41: ………however, commercial breeds could be differ depending on growing region………….. Please fix the grammatical error.
Response: Corrected to “….however, commercial breeds could differ depending on growing region.”
Comment: Lines 42-43: It was found significant genetic variance………….. Please change it to- Significant genetic variation was found ……………..
Response: Corrected to “Significant genetic variation was found in ….”
Comment: Lines 64-65: Please rephrase
Response: Rephrased to “Livni is one of the Russian local pig breeds that is characterized by excellent meat and fat quality.”
Comment: Line 127: Flurometer was used to check ‘purity’ of DNA? OR concentration?
Response: Corrected to «Concentrations of dsDNA solutions were determined using a Qubit 1.0 fluorometer (Invitrogen, Life Technologies, USA)»
Comment: Line 133: Large White and Landrace… Already abbreviated, use abbreviation.
Response: Large White and Landrace were replaced be abbreviations (LW and LN)
Comment: Line 155: ……….ancestral populations (K) from one to five…………How was this range determined? Results says the range of one to eight. Please check it.
Response: Corrected to «…the number of ancestral populations (k) from one to eight»
Comment: Be consistent in writing the term FST.
Response: Corrected to FST throughout the manuscript as well as the other indices.
Comment: Line 215: CV error- This abbreviation has never been defined prior to this.
Response: The abbreviation was added in subsection 2.3. “A cross-validation (CV) procedure….”
Comment: Line 216: … from one to eight…. Conflicted with methods. Line 155
Response: Corrected to «…the number of ancestral populations (k) from one to eight»
Comment: Tables: Better to use other symbols than 1, 2. For example, n2 looks like square of n.
Response: Numbers were changed into symbols or letters in all Tables.
Comment: Table 5: The details of the gene names?
Response: Details of the gene names presented in Tables S4-S6 in Supplementary Materials. We added the link to the Tables S4-S6 in the manuscript.
Best regards,
Authors.

Reviewer 2 Report
Overall, the manuscript is well structured and clearly written, and deals with a very interesting topic that highlights the importance of protecting local varieties at the expense of those imported from geographic regions with different climates.
The Introduction provides the reader with sufficient background to easily follow the study, which has the primary objective of identifying the genomic regions and genes potentially involved in the human-mediated selection process.
The Mat&Met and Results sections are detailed and concise. The issue of possible admixture in Livni pigs has been tackled in a rigorous and timely manner, as well as the identification of an artificial selection signature that was found to be more evident in commercial breeds than in Livni pigs.
In my opinion, there are two additional points that could be analysed to furtherly improve the paper: the estimation of any potential inbreeding and the kinship level within each breed.
Here is the list of my comments and suggestions:
- Line 42: ….however, commercial breeds could 41 be differ depending on growing region.
Delete “be”
- Lines 46-48: “Random genetic drift……than in Chinese Duroc”
In the previous paragraph you stated that “The Chinese Duroc was found to have lost a considerable amount of genetic diversity over the past decade, while Chinese Landrace was found to have kept a high genetic diversity”.
Having the above in mind, I would expect that the impact on the loss of genetic diversity was more obvious in Chinese Duroc than in Chinese Landrace, instead you argue the opposite. Please, clarify this point.
- Line 77: I would suggest to add “and integrity” after genetic diversity
- Line 78: “breed-it” is probably a typo, check it
- Line 84: Please replace highest with higher
- Lines 91-92: you don’t need to provide with an anticipation of the results here. Please, move the last sentence to the result section or delete it.
- Line 229: “The discovered candidate genes were found on seven autosomes…..” Please check the number of autosomes carefully, as the one indicated appears to be incorrect.
- Line 349: “This may be a consequence of the participation of various breeds of…..” Remove “of” after “breeds”
Have you considered the possibility of inbreeding in commercial breeds? it is possible that the reduced genetic variability that you detected in the commercial breeds is also due to that. It is quite plausible that the selection of traits leads to a nucleus of strongly related individuals. Furthermore, I would suggest to carry out an analysis of the relatedness levels within each group, it could be useful to have a more detailed picture and new elements to discuss.
- Line 463: delete the comma between genes and which
- Line 484: delete the comma between genes and which
- Line 520: delete the comma between genes and associated
Author Response
We would like to thank the Reviewer 2 for time spent carefully reading of our article. All the given comments, which have been very helpful in improving our manuscript, have been taken into account. Please, find below our response to the comments. We marked all of changes in the manuscript by red.
Comment: Overall, the manuscript is well structured and clearly written, and deals with a very interesting topic that highlights the importance of protecting local varieties at the expense of those imported from geographic regions with different climates.
The Introduction provides the reader with sufficient background to easily follow the study, which has the primary objective of identifying the genomic regions and genes potentially involved in the human-mediated selection process.
The Mat&Met and Results sections are detailed and concise. The issue of possible admixture in Livni pigs has been tackled in a rigorous and timely manner, as well as the identification of an artificial selection signature that was found to be more evident in commercial breeds than in Livni pigs.
Response: Dear Reviewer 2, we are very grateful for your positive feedback on our work. Based on your suggestions we made corrections in the text of the manuscript.
Comment: In my opinion, there are two additional points that could be analysed to furtherly improve the paper: the estimation of any potential inbreeding and the kinship level within each breed.
Response: we evaluated the kinship level by calculating kinship (relatedness) coefficient based on pair-wise identity by state matrix. For Livni pigs, the threshold for relatedness coefficient was set on 0.45. All animals of Livni breed passed through the quality control step for the kinship. We did not apply the kinship filter for commercial breeds. We did not mention it in the first draft of our manuscript as considered these data as not significant. We included this in the revised version of our manuscript (see section 2.2. Quality Control).
Additionally, we calculated individual genomic inbreeding values based on ROH (FROH). We provided the FROH values for each individual in the supplementary materials, Table S10. The mean values of FROH for each breed are provided in Tables S1.
Comment: Line 42: ….however, commercial breeds could 41 be differ depending on growing region.
Delete “be”
Response: we made correction
Comment: Lines 46-48: “Random genetic drift……than in Chinese Duroc”
In the previous paragraph you stated that “The Chinese Duroc was found to have lost a considerable amount of genetic diversity over the past decade, while Chinese Landrace was found to have kept a high genetic diversity”.
Having the above in mind, I would expect that the impact on the loss of genetic diversity was more obvious in Chinese Duroc than in Chinese Landrace, instead you argue the opposite. Please, clarify this point.
Response: we corrected the paragraph by deleting the sentence: “Random genetic drift also showed substantial impact on the loss of genetic diversity, which is more obvious in Chinese Landrace than in Chinese Duroc”.
Comment: Line 77: I would suggest to add “and integrity” after genetic diversity
Response: we made correction.
Comment: Line 78: “breed-it” is probably a typo, check it
Response: A typo was corrected.
Comment: Line 84: Please replace with higher
Response: we made correction.
Comment: Lines 91-92: you don’t need to provide with an anticipation of the results here. Please, move the last sentence to the result section or delete it.
Response: we made correction by deleting the sentence.
Comment: Line 229: “The discovered candidate genes were found on seven autosomes…..” Please check the number of autosomes carefully, as the one indicated appears to be incorrect.
Response: The number of autosomes was corrected on eight.
Comment: Line 349: “This may be a consequence of the participation of various breeds of…..” Remove
Response: we made correction
Comment: Have you considered the possibility of inbreeding in commercial breeds? it is possible that the reduced genetic variability that you detected in the commercial breeds is also due to that. It is quite plausible that the selection of traits leads to a nucleus of strongly related individuals. Furthermore, I would suggest to carry out an analysis of the relatedness levels within each group, it could be useful to have a more detailed picture and new elements to discuss.
Response: We selected the animals for our studies considering the pedigrees. Additionally, we calculated the kinship (relatedness) coefficient based on pair-wise identity by state matrix. For Livni pigs, the threshold for relatedness coefficient was set on 0.45. All animals of Livni breed passed through the quality control step for the kinship. We did not apply the kinship filter for commercial breeds. We did not mention it in the first draft of our manuscript as considered these data as not significant. We included this in the revised version of our manuscript (see section 2.2. Quality Control).
Comment: Line 463: delete the comma between genes and which
Response: we made correction
Comment: Line 484: delete the comma between genes and which
Response: we made correction
Comment: Line 520: delete the comma between genes and associated
Response: we made correction
Best regards,
Authors.

Round 2
Reviewer 1 Report
Thank you for incorporating the comments and suggestions. All the very best!
Author Response
Dear Reviewer,
We are very grateful for your positive feedback on our revised manuscript.
Best regards,
Authors.